# Application of Thyroid Hormones in Women’s Hair for the Non-Invasive Prediction of Graves’ Disease

**DOI:** 10.3390/biom15030353

**Published:** 2025-02-28

**Authors:** Kouhei Igarashi, Chie Takita, Masako Matsumoto, Wataru Kitagawa, Atsuko Ota, Naoko Miyazaki, Koichi Ito, Kazutaka Ikeda

**Affiliations:** 1Laboratory of Biomolecule Analysis, Department of Applied Genomics, Kazusa DNA Research Institute, 2-6-7 Kazusa Kamatari, Kisarazu 292-0818, Japan; kouhei.igarashi@aderans.com (K.I.); chie.takita@aderans.com (C.T.); 2R&D Office, Aderans Co., Ltd., 1-26-6 Shinjuku, Shinjuku-ku, Tokyo 160-0022, Japan; 3Department of Internal Medicine, Ito Hospital, 4-3-6 Jingu-mae, Shibuya-ku, Tokyo 150-8308, Japan; m-matsumoto@ito-hospital.jp; 4Department of Surgery, Ito Hospital, 4-3-6 Jingu-mae, Shibuya-ku, Tokyo 150-8308, Japan; w-kitagawa@ito-hospital.jp (W.K.); k-ito@ito-hospital.jp (K.I.); 5Clinical Laboratory Division, Ito Hospital, 4-3-6 Jingu-mae, Shibuya-ku, Tokyo 150-8308, Japan; n-miyazaki@ito-hospital.jp; 6Laboratory of Omics and Informatics, Department of Molecular and Chemical Life Sciences, Graduate School of Life Sciences, Tohoku University, 2-1-1 Katahira, Aoba-ku, Sendai 980-8577, Japan

**Keywords:** thyroid hormone, Graves’ disease, non-invasive hair screening, lipidomics, mass spectrometry, health check

## Abstract

Graves’ disease (GD) is an autoimmune disorder that can be difficult to distinguish from other diseases due to symptom similarity. The exacerbation of GD owing to delayed diagnosis is a serious issue, and a novel accessible health screening system is needed. Therefore, this study investigated the association between GD and thyroid hormone levels in women’s hair and evaluated the prediction accuracy of this non-invasive type of sample. By optimizing pretreatment and analysis techniques using liquid chromatography–mass spectrometry (LC-MS), free triiodothyronine (FT3) and thyroxine (FT4) could be detected in only 2 mg of hair with high sensitivity. Compared with healthy controls, the thyroid hormone levels in the hair of GD patients were significantly higher in correlation with blood levels. The predictive ability of hair thyroid hormones was analyzed using a receiver operating characteristic (ROC) curve, and the optimal cut-off value was determined via the Youden index. As a result, the area under the curve (AUC) was 0.974 (95% confidence interval (CI): 0.935–1.000) for FT3 and 0.900 (95% CI: 0.807–0.993) for FT4. The cut-off value was 0.133 pg/mg (sensitivity: 91.2%; specificity: 100%; positive predictive value (PPV): 100%; negative predictive value (NPV): 76.9%) for FT3 and 0.067 pg/mg (sensitivity: 70.6%; specificity: 100%; PPV: 100%; NPV: 50.0%) for FT4. Collectively, our new approach offers the possibility of accurately and non-invasively detecting GD using hair samples. Since hair can be stored and transported at room temperature, this system facilitates large-scale screening at locations including hair salons and homes, potentially enabling the early determination of GD outside of medical facilities.

## 1. Introduction

Thyroid hormones (triiodothyronine [T3] and thyroxine [T4]) are synthesized from the L-tyrosine residue of thyroglobulin by the follicular cells of the thyroid gland through the action of thyroid peroxidase (TPO) (Figure 1). Once produced, these hormones circulate in the blood and are primarily bound to thyroid hormone-binding proteins. A fraction of these hormones exist as free active thyroid hormones (free T3 [FT3] and free T4 [FT4]), which are critical for the regulation of various biological functions, including energy metabolism and autonomic nervous system activities [1,2,3]. The secretion of thyroid hormones is regulated by thyroid-stimulating hormone (TSH), which is produced by the pituitary gland in the brain, ensuring that their blood concentrations are maintained within the appropriate range [4].

Graves’ disease (GD) and Hashimoto’s disease (HD) are widely recognized autoimmune conditions that lead to thyroid hormone imbalances [5]. GD in particular is notable for its symptoms, such as thyroid enlargement, tachycardia, and exophthalmia, which result from the hypersecretion of thyroid hormones due to increased thyroid function driven by TSH receptor antibody (TRAb) [6]. Consequently, blood tests for GD primarily check for elevated levels of FT3 and FT4, low levels of TSH, and increased TRAb and thyroid-stimulating antibody (TSAb) levels [7].

The main characteristics of GD include its higher prevalence in women [8] and the difficulty of distinguishing it from other diseases, even when symptoms are present [9]. For example, menopausal symptoms such as tachycardia and sweating are difficult to distinguish from GD [10]. Consequently, the delayed diagnosis of GD has become a significant health concern, leading to a decline in quality of life [11,12,13]. Moreover, GD is associated with an increased risk of atrial fibrillation or fetal loss and severe conditions such as thyrotoxic periodic paralysis (TPP) [14,15,16,17]. In light of these challenges, there is an urgent need for the early detection and treatment of GD. Therefore, developing a new GD assessment system, besides the existing medical testing protocols, is essential.

This study focused on the trace amounts of thyroid hormones present in human hair [18,19] and evaluated whether they could be used to predict GD. The usefulness of hair in this approach is supported by its relative stability compared to blood, as it is less susceptible to diurnal and day-to-day fluctuations. Hair grows at a rate of approximately 1 cm per month, serving as “memory tissue” that records past physiological states, making it a valuable tool for retrospective analysis. Indeed, hair has been effectively utilized for drug testing and assessments of exposure to toxic heavy metals such as lead, arsenic, and mercury [20,21]. In addition, hair collection is non-invasive compared with blood sampling, and hair samples can be transported easily and stored at room temperature. These attributes suggest the potential to integrate a GD assessment system into daily life beyond medical facilities, including at hair salons and in homes.

Human hair contains trace amounts of thyroid hormones, but conventional detection methods typically require a large quantity of hair. Thus, greater sensitivity is needed to alleviate the burden of sampling on subjects. Moreover, the association between GD and thyroid hormone levels in women’s hair remains unclear.

To address these challenges, a highly sensitive liquid chromatography–mass spectrometry (LC-MS) analysis system was developed to detect FT3 and FT4 in hair samples. This study evaluated the correlation between the levels of these hormones in the hair and blood of individuals with GD while also assessing the prediction accuracy of using this non-invasive type of sample.

The L-tyrosine residue of thyroglobulin present in the follicular cells of the thyroid gland is iodinated by the action of thyroid peroxidase (TPO) to form monoiodotyrosine (MIT) and diiodotyrosine (DIT). When MIT and DIT combine, triiodothyronine (T3) is generated, and when two DIT molecules combine, hyroxine (T4) is generated. This is a simplified overview of the structure and biosynthesis pathway of thyroid hormones and their related compounds.

## 2. Materials and Methods

### 2.1. Study Design

This study was approved by the Ethics Review Board of Ito Hospital (approval number: 352) and was conducted in accordance with the Declaration of Helsinki. Written informed consent was obtained from all participants. It involved Japanese patients with GD who were either newly diagnosed or were experiencing disease recurrence, as well as healthy control (HCT) volunteers. HCT volunteers were identified through blood tests that ensured the following values fell within normal ranges: FT3, 2.2–4.3 pg/mL; FT4, 0.80–1.60 ng/dL; TSH, 0.20–4.50 µIU/mL; thyroglobulin antibody (TgAb), ≤40 IU/mL; and human thyroglobulin (HTg), ≤33.7 ng/mL. Physicians diagnosed GD based on test values at the time of initial diagnosis or recurrence, where TSH levels were lower than the normal range, and FT3, FT4, or both were also elevated. The sample population comprised 34 women in the GD group and 10 women in the HCT group (Table 1). Using scissors, hair samples were collected from the base of the scalp of each participant with GD at a position corresponding to the period of disease prior to treatment. Samples were then stored at room temperature.

### 2.2. Reagents and Standards

Methanol, distilled water, and acetonitrile for LC-MS were purchased from Fujifilm Wako Pure Chemical Industries, Ltd. (Osaka, Japan). The standard compounds, 3,3′,5-Triiodo-L-thyronine (T3) and L-Thyroxine (T4) were obtained from Sigma-Aldrich (St. Louis, MO, USA), and 3,3′,5-Triiodo-L-thyronine-13C6 (13C6-T3) and L-Thyroxine-13C6 (13C6-T4) were obtained from Cerilliant (Round Rock, TX, USA).

### 2.3. Hair Sample Preparation

The collected hair samples were placed in a 50 mm disposable dish designated for washing. To initiate the washing process, 10 mL of distilled water, which was heated in a constant-temperature bath at 40 °C, was added to the dish. After shaking the dish for 5 min, the distilled water was carefully removed. This washing process was repeated twice. Subsequently, 10 mL of room-temperature methanol was added to the dish. Following another 5 min of shaking, the methanol was removed, and the dish was left to dry overnight at room temperature with the lid left semi-open. After drying and cutting the hair samples to 1 cm from the root, they were individually weighed to 2 mg and transferred into inert glass inserts (Agilent, Santa Clara, CA, USA). Subsequently, 100 µL of a methanol/distilled water (1/1, *v*/*v*) solution containing internal standards (13C6-T3: 16.7 pM; 13C6-T4: 13.3 pM) was added to each hair extract. The mixture was shaken at 300 rpm for 4 h at 60 °C. Following this, the dehaired supernatant was carefully transferred into a new inert glass insert. Finally, 115 µL of methanol/distilled water (9/14, *v*/*v*) was added to the supernatant as a diluent.

To isolate the thyroid hormone from the extract, a trace amount of solid phase extraction was conducted using a silica gel-based reverse-phase octadecylsilyl (C18) column (AiSTI SCIENCE, Wakayama, Japan). Initially, the column was conditioned once with 250 µL of methanol followed by 250 µL of distilled water. The extract was then loaded onto the column and allowed to interact with the stationary phase. After the loading step, the target analytes were eluted using 200 µL of acetonitrile/methanol/distilled water (6/3/1, *v*/*v*/*v*). The eluate containing the purified thyroid hormone was collected into an inert glass insert. Next, this eluate was subjected to drying using a centrifugal concentrator (Labconco, Kansas City, MO, USA) to remove any residual solvent. Once dried, 60 µL of methanol/distilled water (1/1, *v*/*v*) was added to reconstitute the sample. The reconstituted sample was then subjected to vortexing for 1 min, followed by sonication for 1 min to ensure homogeneity and prepare the sample for LC-MS analysis.

### 2.4. Blood Tests

Blood samples measuring 5 mL were collected and centrifuged (H-60R; Kokusan Co., Ltd., Saitama, Japan) at room temperature (20 ± 8 °C) and 3000 rpm for 7 min. Following centrifugation, 500 µL of serum was carefully extracted from each sample. These serum samples were then analyzed using the Eclusis reagent protocol. Specifically, the following volumes of serum were used to obtain measurements: 9 µL for FT3 and FT4, 30 µL each for TSH and TRAb, 6 µL for TgAb, 12 µL for TPOAb, and 21 µL for HTg. For the quantification of FT3, FT4, and TSH concentrations, electrochemiluminescence immunoassay was employed with a cobas analyzer (Roche Diagnostics, Basel, CHE).

### 2.5. LC-MS Analysis

LC-MS analysis was applied to the thyroid hormones using a triple-quadrupole mass spectrometer (LCMS-8060; Shimadzu, Kyoto, Japan) equipped with an ultra-performance liquid chromatography (Nexera X2; Shimadzu). LC separation was performed with a gradient elution of mobile phases A (water) and B (methanol). The gradient conditions were as follows: 20% B, 0–2 min; 20–98% B, 2–7 min; 98% B, 7–9 min; 98–20% and 20% B, 9–14 min. The flow rate was 300 µL/minute at 40 °C using a Kinetex C8 (100 × 2.0 mm i.d., particle size 1.7 µm; (Phenomenex, Torrance, CA, USA), and the injection volume was 50 µL in positive-ion mode. The MS conditions were set as follows: nebulizer gas flow, 2.5 L/minute; heating gas flow, 10 L/minute; drying gas flow, 10 L/minute; heat block temperature, 350 °C; desolvation line temperature, 650 °C; and spray voltage, 4.0 kV for positive-ion mode. Multiple-reaction-monitoring (MRM) mode was applied to all targeted thyroid hormones. Details of the optimized MRM parameters are presented in Table 2.

### 2.6. Data Analysis

MRM data analysis was performed using LabSolutions (Shimadzu) according to the Level 1 criteria of the Lipidomics Standard Initiative (LSI) guidelines [22,23]. As a quantitative approach, peak areas were computed using T3 and T4 as external standard compounds with known concentrations to establish a calibration curve. Molar concentrations were then derived from the peak areas obtained from the sample measurements. Subsequently, the molar concentrations were converted to calculate the quantity of FT3 and FT4 present in 1 mg of hair (pg/mg). Statistical analyses were performed using JMP software 17.2.0 (SAS Institute, Cary, NC, USA). Student’s *t*-test was utilized to assess significant differences, while the area under the receiver operating characteristic (ROC) curve (AUC) was employed to evaluate the accuracy of the prediction. Moreover, the Youden index, which indicates optimal sensitivity and specificity, provided the cut-off value for determination.

## 3. Results

### 3.1. Establishment of Highly Sensitive System for Analyzing Thyroid Hormones in Hair

In this study, new LC-MS pretreatment and separation analysis methods were investigated to enable highly sensitive thyroid hormone detection using as little hair as possible to reduce the burden of sampling on subjects. A reversed-phase trace solid phase extraction method was applied for pretreatment. Thyroid hormones were eluted from the hair samples with 100 µL of 50% methanol solution, and the extract was purified, using a trace solid-phase column with C18 as a carrier, and dried. The thyroid hormone concentration was then enhanced by redissolving the extract with 60 µL of 50% methanol solution. Additionally, during LC separation, a mixed gradient (mobile phase A: water; mobile phase B: methanol) with less background noise was used. To reduce sample loss during injection into the LC as much as possible, the composition of the initial mobile phase of the LC was changed to a 20% methanol solution to stabilize adsorption onto the LC column.

As a result, it became possible to perform stable analysis without distorting the peak shape even after injecting 50 µL, which is approximately 83% of the 60 µL volume of re-dissolved solution. Furthermore, in the MS analysis, selective analysis of thyroid hormones was performed using triple-quadrupole MS in MRM mode (Table 2). As a result, the final LC-MS quantification for FT3 ranged from 10 pM to 10 nM (limit of detection [LOD]: 1 pM), and the FT4 quantification ranged from 25 pM to 10 nM (LOD: 10 pM). Therefore, it was possible to detect FT3 and FT4 from only 2 mg of hair with high sensitivity (Table 3).

### 3.2. Evaluation of Accuracy of GD Determination Using Thyroid Hormones in Hair

Using the highly sensitive analysis system constructed in this study, we examined the levels of FT3 and FT4 in the hair of individuals diagnosed with GD based on blood tests. Our findings revealed that the concentrations of FT3 (*p* < 0.001) and FT4 (*p* < 0.01) in the hair of the GD patients were significantly elevated compared to those in the HCT volunteers (Figure 2). In addition, the AUC values for FT3 and FT4 were 0.974 (95% CI: 0.935–1.000) and 0.900 (95% CI: 0.807–0.993), respectively, indicating a high level of determination accuracy (Figure 3).

Furthermore, the Youden index, which indicates the optimal balance between sensitivity and specificity, was utilized to establish the cut-off value. As a result, the cut-off value was 0.133 pg/mg (sensitivity: 91.2%; specificity: 100%; positive predictive value (PPV): 100%; negative predictive value (NPV): 76.9%) for FT3 and 0.067 pg/mg (sensitivity: 70.6%; specificity: 100%; PPV: 100%; NPV: 50.0%) for FT4 (Figure 4). These results suggest that FT3 is more suitable for predicting GD in hair due to a low rate of false negatives.

### 3.3. Verification of Correlation Between Blood and Hair Tests for GD

To validate the quantitative relationship between the levels of FT3 and FT4 in hair and their concentration in blood, we analyzed the correlation coefficient using scatter plots. Remarkably, a positive correlation was found between each parameter, with FT3 exhibiting a particularly strong correlation (*r* = 0.828) in comparison to FT4 (*r* = 0.601) (Figure 5). Notably, a quantitative correlation between the levels of thyroid hormones in hair and their concentration in blood was observed.

## 4. Discussion

The method of detecting thyroid hormones in hair typically involves the highly sensitive LC-MS detection technique [18]. However, due to the minute quantity of thyroid hormones present in hair, conventional methods necessitate approximately 7.5 to 50 mg of hair for hormone detection [18,19]. This imposes a significant burden on the subjects.

However, through the optimization of extraction and separation techniques in this study, we were able to analyze the thyroid hormone content using only 2 mg of hair, approximately one-quarter of the minimum quantity previously required. This substantial reduction in sample size considerably alleviates the burden on subjects. Moving forward, we aim to further enhance the sensitivity of our LC-MS analysis system. As it enables the detection of thyroid hormone levels from just a few strands of hair, our method has the potential to popularize hair-based health assessments.

To the best of our knowledge, this study is the first to demonstrate the potential to detecting GD based on FT3 and FT4 levels in hair. Notably, a quantitative correlation Ibetween the levels of thyroid hormones in hair and their concentration in blood was observed. This suggests a mechanism wherein thyroid hormones, akin to minerals and cystine, are absorbed from the bloodstream into hair matrix cells, where they are incorporated into the growing hair shaft [24].

This study has some limitations in terms of prediction accuracy. Although we were able to detect GD in all patients based on their blood thyroid hormone levels, it is also known that the thyroid-stimulating immunoglobulin (TSI) assay has a very high sensitivity (96%) and specificity (99%) for the blood diagnosis of GD [25]. Considering this information, the predictive accuracy of the hair assay needs to be further improved for practical future use, though our preliminary data suggest a trend toward increased accuracy when FT3 and FT4 levels in hair are plotted on two axes and both cutoff values are applied (Appendix A). Moreover, as we exclusively utilized hair samples from Japanese women in this initial study, our results may not be generalizable. To promote the global adoption of hair screening for GD, it is imperative to validate the reproducibility of our findings using hair samples from patients with diverse racial backgrounds and expand our dataset to enhance prediction accuracy.

This study may bring us closer to realizing an accessible and non-invasive health assessment tool that differs from conventional blood tests performed at medical institutions. This may lead to the early detection of many latent cases of GD in everyday settings. Furthermore, considering that hair grows approximately 1 cm per month and accumulates metabolites, this approach may create opportunities for longitudinal health monitoring capable of retracing past states with a single hair sample. By extending such long-term data fluctuation assessments to marker molecules associated with various other diseases, such as menopausal disorder, we may move closer to achieving straightforward personalized healthcare solutions.

## 5. Conclusions

In this study, we developed a highly sensitive LC-MS analysis system that can detect FT3 and FT4 using only 2 mg of hair. Using this optimized method, we confirmed that the concentrations of FT3 and FT4 in the hair of GD patients were significantly elevated compared with those in HCT volunteers. We also validated the predictive ability of thyroid hormone levels in the hair of GD patients via an ROC curve and the maximum Youden index value. The AUC values for FT3 and FT4 were 0.974 (95% CI: 0.935–1.000; sensitivity: 91.2%; specificity: 100%; PPV: 100%; NPV: 76.9%) and 0.900 (95% CI: 0.807–0.993; sensitivity: 70.6%; specificity: 100%; PPV: 100%; NPV: 50.0%), respectively. Furthermore, we observed a positive correlation between the levels of FT3 and FT4 in hair and their concentration in blood.

Our new approach offers the possibility to non-invasively and accurately detect GD using hair samples. Since hair can be stored and transported at room temperature, this system facilitates large-scale screening in locations such as hair salons and in-homes, potentially enabling the early detection of GD outside of medical facilities.

## Figures and Tables

**Figure 1 biomolecules-15-00353-f001:**
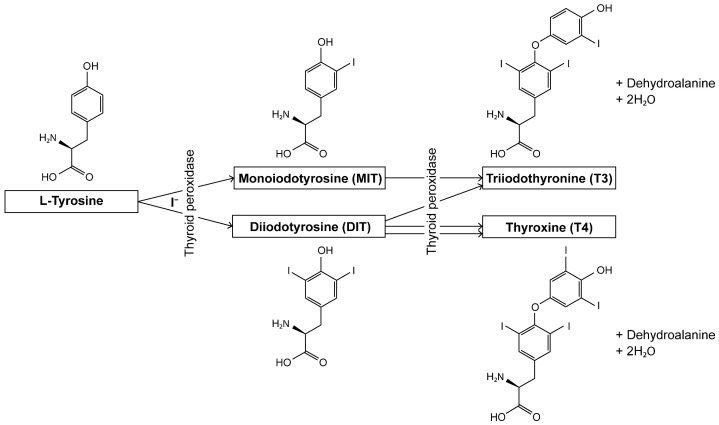
Thyroid hormone biosynthesis pathway.

**Figure 2 biomolecules-15-00353-f002:**
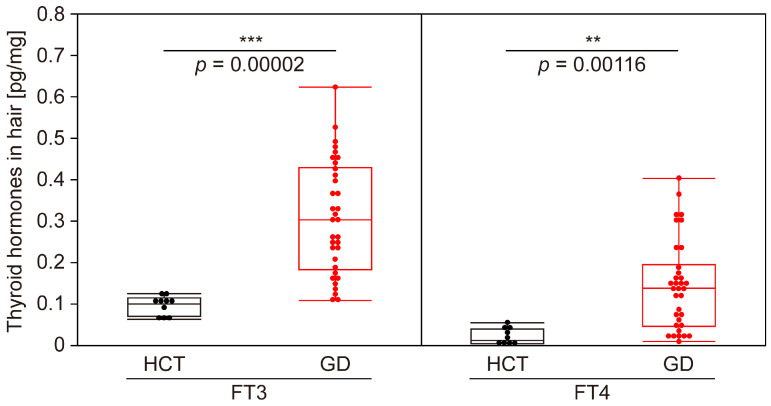
Comparison of thyroid hormone levels in the hair of GD patients and HCT volunteers. A box-plot diagram comparing the amounts of FT3 and FT4 (pg/mg) in the hair of HCT volunteers (*n* = 10) and GD patients (*n* = 34). The top and bottom of the box indicate the 75th and 25th percentiles, respectively; the line within the box indicates the median, and the bars indicate the maximum and minimum values. ** *p* < 0.01; *** *p* < 0.001.

**Figure 3 biomolecules-15-00353-f003:**
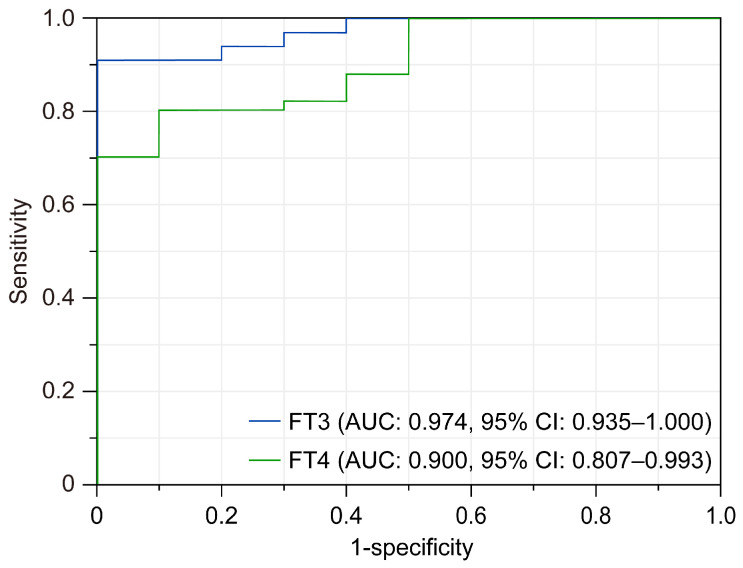
Evaluation of GD determination accuracy via ROC curve analysis. Receiver operating characteristics (ROC) curves for FT3 (blue line) and FT4 (green line) in hair are shown. Area under the curve (AUC) values are 0.974 (95% confidence interval (CI): 0.935–1.000) and 0.900 (95% CI: 0.807–0.993) for FT3 and FT4, respectively.

**Figure 4 biomolecules-15-00353-f004:**
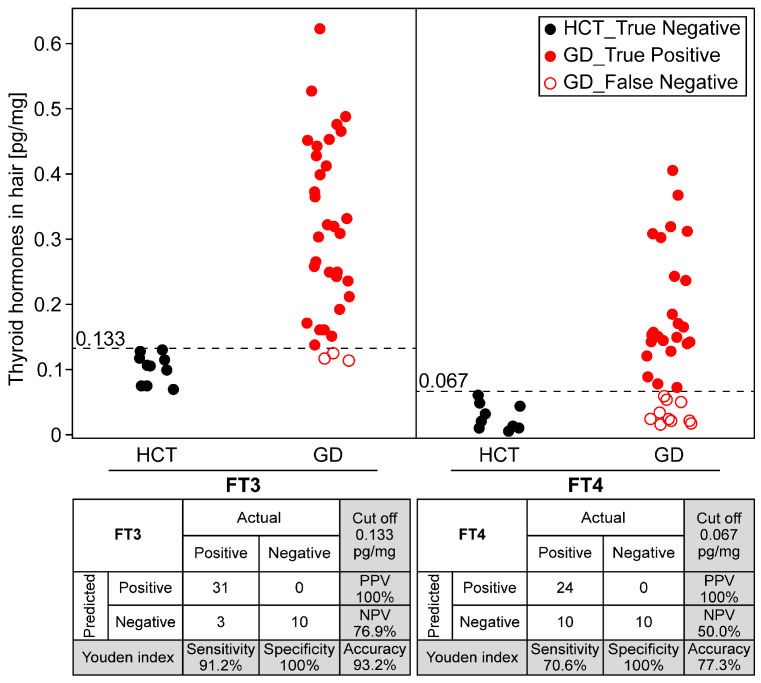
Predictive determination of GD using thyroid hormones in hair. A plot diagram showing the amount of FT3 and FT4 in hair of HCT volunteers (true negative: filled black circle) and GD patients (true positive: filled red circle; false negative: empty red circle). The cut-off value (dashed line) was 0.133 pg/mg (sensitivity: 91.2%; specificity: 100%; positive predictive value (PPV): 100%; negative predictive value (NPV): 76.9%) for FT3 and 0.067 pg/mg (sensitivity: 70.6%; specificity: 100%; PPV: 100%; NPV: 50.0%) for FT4.

**Figure 5 biomolecules-15-00353-f005:**
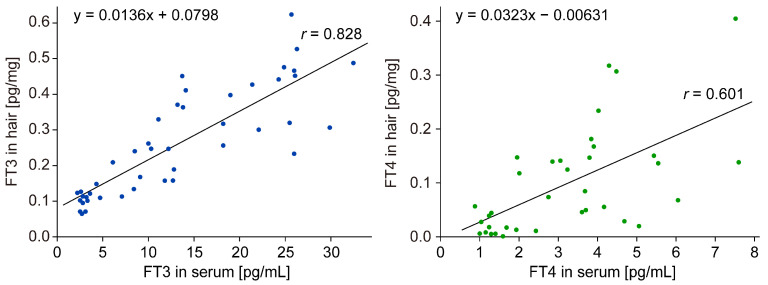
Quantitative correlation of thyroid hormones in serum and hair. Scatter plots of FT3 (filled blue circle) and FT4 (filled green circle), with the amount in serum on the x axis and the amount in hair on the y axis. A positive correlation was observed for both FT3 and FT4, with correlation coefficients of 0.828 and 0.601, respectively.

**Table 1 biomolecules-15-00353-t001:** Age and blood test data of the participants.

	HCT	GD
Number	10	34
Age (mean ± SD, median, min–max) [years]	36.8 ± 7.03, 39, 26–50	41.7 ± 10.24, 43, 20–59
FT3 (median, min–max) [pg/mL]	2.8, 2.2–3.3	13.8, 3.6–32.5 ≤ 3
FT3 abnormal rate	0 (normal:abnormal = 10:0)	0.941 (normal:abnormal = 2:32)
FT4 (median, min–max) [ng/dL]	1.25, 0.89–1.60	4.24, 1.69–7.77 ≤ 3
FT4 abnormal rate	0 (normal:abnormal = 10:0)	1 (normal:abnormal = 0:34)
TSH (median, min–max) [μIU/mL]	1.58, 0.72–3.35	≤30.01, ≤30.01–0.02
TgAb (mean ± SD, median, min–max) [IU/mL]	12.8 ± 1.40, 12.5, 10.7–15.2	
HTg (mean ± SD, median, min–max) [ng/mL]	9.89 ± 4.52, 8.88, 3.46–17.90	

Abbreviations: HCT, healthy control; GD, Graves’ disease; FT3, free triiodothyronine; FT4, free thyroxine; TSH, thyroid-stimulating hormone; TgAb, thyroglobulin antibody; HTg, human thyroglobulin; SD, standard deviation.

**Table 2 biomolecules-15-00353-t002:** Mass spectrometry parameters of thyroid hormones.

Compound Name	Molecular Formula	Monoisotopic Mass	Precursor ion (Adduct)	Transition of Confirmation	CE (V)	Transition of Quantification	CE (V)
Triiodothyronine (T3)	C15H12I3NO4	650.7901(656.8102) *^1^	[M + H]^+^	651.80 > 651.80	−9.0	651.80 > 605.80	−23.0
				(657.90 > 657.90) *^1^	−9.0	(657.90 > 611.90) *^1^	−23.0
Thyroxine (T4)	C15H11I4NO4	776.6867(782.7068) *^2^	[M + H]^+^	777.70 > 777.70	−14.0	777.70 > 731.60	−25.0
				(783.80 > 783.80) *^2^	−14.0	(783.80 > 737.80) *^2^	−25.0

*^1^: Triiodothyronine-13C6, *^2^: Thyroxine-13C6. Abbreviations: CE, collision energy.

**Table 3 biomolecules-15-00353-t003:** Quantitative evaluation of thyroid hormones.

	RT RSD (%, *n* = 5)				RPA RSD (%, *n* = 5)							
Compound Name	Day 1	Day 2	Day 3	Inter Day	Day 1	Day 2	Day 3	Inter Day	LOD (pM)	Linearity Range (pm)	Linearity (R^2^)	Recovery (%)
T3	0.07	0.04	0.05	0.05	1.53	6.49	5.78	3.09	1.0	10–1000 (Low) *^1^	0.9962	91.6
										1000–10,000 (High) *^2^	0.9997	
T4	0.03	0.05	0.04	0.03	3.02	3.50	2.81	3.22	10.0	25–1000 (Low) *^1^	0.9971	92.3
										1000–10,000 (High) *^2^	0.9982	

*^1^, *^2^ each calibration curve was used separately for quantification to cope with a wide range of concentrations. Abbreviations: LOD, limit of detection; RPA, relative peak area; RSD, relative standard deviation; RT, retention time.

## Data Availability

The datasets used and/or analyzed in this study are available from the corresponding author upon reasonable request.

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
