# Peer review of "Application of Thyroid Hormones in Women’s Hair for the Non-Invasive Prediction of Graves’ Disease"

_biomolecules, 2025, doi:10.3390/biom15030353_

Round 1
Reviewer 1 Report
Comments and Suggestions for Authors
In this study, the authors develop an LC-MS method to detect free T3 and free T4 in hair samples with the goal of detecting Graves’ disease. The T3/T4 measurements in hair were compared to those in blood and could be used to predict whether the patient had Graves’ disease.
A few minor comments/questions:
What is the LOD of the standard clinical blood method for T3 and T4?
Do the FT3 and FT4 values in your hair samples correlate? Relatedly, are the 3 samples below the cut-off for FT3 also below the cut-off for FT4?
What steps would need to be taken to make the prep of samples high throughput and possible in a clinical setting?
Are there any related thyroid hormone metabolites or proteins (like TSH) that could be detected in hair that could improve the predictive ability of this strategy?
Why are TgAb and HTg measured for the healthy but not GD samples? What are the implications of those values?
Comments on the Quality of English LanguageIn general, the quality of the writing was quite good, but there are several spots in the abstract that could be improved.
Author Response
Reviewer â‘
(Comments)
In this study, the authors develop an LC-MS method to detect free T3 and free T4 in hair samples with the goal of detecting Graves’ disease. The T3/T4 measurements in hair were compared to those in blood and could be used to predict whether the patient had Graves’ disease.
(Response)
We appreciate the time and effort for your review of our paper in detail. We have responded to your comments and questions below and look forward to your consideration.
(Comments)
A few minor comments/questions:
- What is the LOD of the standard clinical blood method for T3 and T4?
(Response)
Thank you for providing this insight. According to the information on the thyroid hormone Eclusis reagent (Roche) used in this study, the LOD and LOQ are as follows.
FT3
LOD: 0.391 pg/mL, LOQ: 0.977 pg/mL
FT4
LOD: 0.391 pg/mL, LOQ: 0.977 pg/mL
Unfortunately, we have not measured thyroid hormones in hair sample using this kit, and we are unable to compare the sensitivity with LC-MS at this stage.
②-1 Do the FT3 and FT4 values in your hair samples correlate?
(Response)
Thank you for your valuable comment. We confirmed the quantitative correlation between FT3 and FT4 ​​in hair, and obtained a high value of 0.755712 (=r). It is known that these hormones ​​correlate in the blood of GD patients, suggesting that the blood profile may also be reflected in hair.
②-2 Relatedly, are the 3 samples below the cut-off for FT3 also below the cut-off for FT4?
(Response)
When we examined the three indicated individuals, two had FT4 levels below the cutoff value and one above the cutoff value.
③ What steps would need to be taken to make the prep of samples high throughput and possible in a clinical setting?
(Response)
Thank you for your very important comment. In this study, we manually perform solid-phase purification and concentration/drying in the pretreatment process, and understand the need to increase throughput. We are already working on an automated system for these processes, but at this stage the recovery rate is lower than manual methods, so we would like to improve this in the future and increase throughput.
④ Are there any related thyroid hormone metabolites or proteins (like TSH) that could be detected in hair that could improve the predictive ability of this strategy?
(Response)
Thank you for your excellent suggestions. We have tried to detect TSH by proteome analysis so far, but have not succeeded yet. Also, it has not been detected in other paper reports to date. We believe that we need to combine such antibodies to improve the accuracy of diagnosis in the hair, including Hashimoto's disease, in the future.
⑤ Why are TgAb and HTg measured for the healthy but not GD samples? What are the implications of those values?
(Response)
Thank you for your careful confirmation. GD patients are basically checked for TgAb and HTg at the first visit. In this study, GD patients who relapsed and visited the clinic again other than the first visit were included, and there was no data on TgAb and HTg in this case. Therefore, it was not possible to obtain TgAb and HTg values ​​close to the time of hair collection in all GD patients, and so values ​​for GD are not shown in the table.

Reviewer 2 Report
Comments and Suggestions for Authors
The paper itself is really interesting and well written; I only have a few observations:
- In the introduction it is stated that ‘The main features of GD include its higher prevalence in women than in men, and the difficulty of distinguishing it from other diseases even in the presence of symptoms’, so I would have expected a justification of the difficulty of distinguishing it from other diseases. I think a sentence could be inserted in this regard.
- I would move table 1 to the end of section 2.1 to make the participants' data immediately visible.
- Lines 181–195 appear more like an explanation of techniques than results.
- Figure 6 is referred to as figure s
Author Response
Reviewer â‘¡
(Comments)
The paper itself is really interesting and well written.
(Response)
We appreciate the time and effort for your review of our paper in detail. We have responded to your observations below and look forward to your consideration.
(Comments)
I only have a few observations:
① In the introduction it is stated that ‘The main features of GD include its higher prevalence in women than in men, and the difficulty of distinguishing it from other diseases even in the presence of symptoms’, so I would have expected a justification of the difficulty of distinguishing it from other diseases. I think a sentence could be inserted in this regard.
(Response)
Thank you for your valuable suggestion. To make it easier to understand, we have included in the text (p. 2, lines 61-62) the example of menopausal symptoms, which can be difficult to distinguish from GD.
②  I would move table 1 to the end of section 2.1 to make the participants' data immediately visible.
(Response)
We are sorry for the lack of consideration. Table 1 has been moved to the end of section 2.1.
③ Lines 181–195 appear more like an explanation of techniques than results.
(Response)
Thank you for your important suggestion. As you pointed out, the results include an explanation of techniques, so we have changed the text (p. 5, lines 190-192). Also, we have moved some of it to the discussion (p. 9, lines 273-276) to make it easier to understand.
④ Figure 6 is referred to as figure s
(Response)
Thank you for checking in detail. It seems that Figure S was mistakenly uploaded as Figure. 6 on the Web system. We will upload the final version without any mistakes.
